# Harnessing the Systemic Biology of Functional Decline and Cachexia to Inform more Holistic Therapies for Incurable Cancers

**DOI:** 10.3390/cancers16020360

**Published:** 2024-01-14

**Authors:** Amber Willbanks, Mina Seals, Reem Karmali, Ishan Roy

**Affiliations:** 1Shirley Ryan AbilityLab, Chicago, IL 60611, USA; 2Feinberg School of Medicine, Northwestern University, Chicago, IL 60611, USA; 3Robert H. Lurie Comprehensive Cancer Center, Northwestern University, Chicago, IL 60611, USA; 4Department of Physical Medicine and Rehabilitation, Northwestern University, Chicago, IL 60611, USA

**Keywords:** advanced cancer, rehabilitation, function, exercise, cachexia, fatigue

## Abstract

**Simple Summary:**

Approaches to treat advanced cancer within tumor biology have been largely unsuccessful. This review investigates the biology downstream of the tumor as a potential new avenue for treatment. We draw insight from the mechanisms that lead to sickness and debility, including a decline in physical/cognitive function and muscle wasting, otherwise known as cachexia.

**Abstract:**

Options for treatment of incurable cancer remain scarce and are largely focused on limited therapeutic mechanisms. A new approach specific to advanced cancers is needed to identify new and effective treatments. Morbidity in advanced cancer is driven by functional decline and a number of systemic conditions, including cachexia and fatigue. This review will focus on these clinical concepts, describe our current understanding of their underlying biology, and then propose how future therapeutic strategies, including pharmaceuticals, exercise, and rehabilitation, could target these mechanisms as an alternative route to addressing incurable cancer.

## 1. Introduction

Advanced cancer describes cancers that are locally advanced or metastatic, have limited treatment options, and typically have low survival. These cancers often do not respond to initial treatment and are aggressive in nature. The four most common cancers have a five-year survival rate of 7–32% in the advanced stages, compared with 50–100% in localized tumors [1]. In addition to its impact on mortality, patients with advanced cancer generally experience increased morbidity and disability burden [2,3]. Beyond standard chemotherapy regimens typically also used for localized disease, advanced cancer treatment largely focuses on palliative care with the hope of alleviating symptom burden throughout the remainder of the disease [4,5]. Some targeted interventions do exist for advanced cancers, including chimeric antigen receptor (CAR) T-cell therapy and immune-checkpoint inhibitors; however, survival outcomes often remain bleak, and further investigation of treatments that directly target advanced disease is needed [6,7,8,9,10,11].

A key feature underlying advanced cancer is the associated systemic physiology beyond the tumor. When cancer progresses beyond the local niche, patients with advanced cancer experience numerous systemic sequelae in parallel with primary tumor progression that contribute to their overall decline [12,13,14]. At this point, cancer is a systemic disease not only due to metastatic spread but also due to its effects on the entire body, which is left more susceptible to additional systemic insults, including thrombogenesis, pain, fatigue, cardiopulmonary decline, and cachexia.

Cancer cachexia is a secondary systemic muscle wasting syndrome caused by systemic inflammation and accelerated catabolic metabolism, which occurs in ~50% of all patients with cancer and 85% of those with advanced cancer [15,16,17,18,19,20]. The presence of cachexia negatively impacts quality of life (QoL), treatment success, and survival [16,21]. The progressive loss of muscle mass in cachexia leads to concurrent impairments in overall muscle strength and physical activity, including walking, proximal muscle strength, and standing tolerance [22,23]. However, the effects of cancer cachexia extend beyond muscle [19,24,25,26,27], impacting whole-body physical function and functional independence with activities of daily living [28,29]. In addition, multiple organ systems, including the liver, intestines, and neuroendocrine network, experience dysfunctional biology, implicating cancer cachexia as a proxy for systemic disease in advanced cancer [16,20].

New approaches to targeting advanced cancer that are focused on tumor mechanisms are in development, but there is considerably less focus in the oncologic community on translating treatments for the parallel, systemic advanced cancer mechanisms. Even so, a handful of promising biological and physiological targets linked to extratumoral outcomes, such as cachexia, are emerging. In this review, we will explore the link between morbidity, functional decline, and the systemic biology of advanced cancer. We will underscore some of the specific physiologic mechanisms that tie together these concepts and then describe some of the multi-modal approaches to treating these mechanisms that could be incorporated into a more holistic approach to advanced cancer treatment.

## 2. Functional Decline and Facilitators of Morbidity in Advanced Cancers

Physical functional decline is a common feature of advanced cancer and is defined as a decrease in independence with mobility or the performance of activities of daily living (ADLs), such as walking, dressing, and using the toilet [30,31]. A decrease in functional independence contributes to morbidity and overall survival [32,33,34], with up to a 3-fold increase in mortality in those with cancer [30]. Importantly, a change in function is distinct from a change in physical activity, whereby a change in function is defined as changes in the ability to perform specified tasks, and physical activity is defined by energy expenditure [35]. In studies of patients with advanced cancer, the available measures of physical function are extremely diverse, including the use of true measures of functional independence [28,29,36] and measurements of physical performance that are proxies for functional decline, such as grip strength, the 6 min walk test (6MWT), gait speed, 30 s sit-to-stand [37,38], and timed up and go test (TUG) [30,33,39].

Recent studies suggest that addressing functional decline specifically as a conduit for treating the systemic biology of advanced cancer may reduce the worsening prognosis experienced by this patient population [19,40,41,42]. Herein, we will describe facilitators of morbidity related to advanced cancer and how each contributes to the overall functional capacity of afflicted patients as a measure of both functional assessments and patient-reported outcome measures (PROMs).

Metastatic disease is a key clinical feature of many advanced cancers that results from several factors, including changing transcriptional activity, chromosomal instability, and cell differentiation properties [43,44,45,46]. Once a metastasis is formed, disruption of secondary organ physiology is the most common cause of cancer-related deaths (~90%), impairing systemic function and directly causing organ failure [47,48]. The metastatic process is energetically expensive and distinct from primary tumor biology [49], utilizing critical energy from functional reserves in the body for the dual need of both spread and growth. The depletion of these energy stores exacerbates fatigue, frailty, and decreased functional capacity compared to localized tumors that utilize energy primarily for growth only and therefore have less anabolic demand [50,51]. As a result, patients with metastasis have significantly more self-reported decline in physical function, fatigue, and QoL than those with only primary tumors [52].

Among the first signs of advanced cancer is decreased dietary intake. This can manifest through multiple symptoms, including a measurable loss of appetite (anorexia), pain, lack of interest (anhedonia), fatigue, or aversion to smells and tastes [53,54]. The effects of decreased dietary intake are compounded by malabsorption due to changing gastrointestinal and microbiome physiology. A cascade of negative clinical outcomes can ensue from poor nutrition [55], including weight loss, depression, fatigue, and decreased QoL [56]. More serious conditions are associated with sustained malnutrition, including severe muscle wasting, functional and cognitive decline, and poor survival [53,55,57]. For example, 6MWT scores have been shown to be both an indication of poor survival and closely associated with malnutrition in multiple studies [34,57,58].

Progressive functional decline is cited among the key outcomes of cancer cachexia, but data on the specific relationship between cachexia and function remain imprecise. Most studies focus on physical performance measures, such as hand grip strength, which typically change at the end stages of the disease rather than the longitudinal study of physical functional independence via the course of cachexia development [26,59,60]. However, more recent studies have investigated potentially more accurate measures of functional impairment throughout the course of the disease by utilizing functional independence measures (FIM), more diverse physical performance measures, and function-specific PROMs [28,35,61,62]. The imprecise application of functional assessments is reflective of a larger issue within the cachexia field in which multiple markers can be used for cachexia diagnosis, including overall weight loss, the weight loss grading scale (WLGS), and biochemical markers such as C-reactive protein (CRP) and albumin [63,64,65]. The variety of diagnostic criteria leads to a non-unified population that contains multiple cohorts with differing presentations of decline. These considerations withstanding, cancer cachexia progression of any diagnostic criteria directly contributes to morbidity, frailty, cognitive decline, and overall survival [19,61,66,67,68]. In ~20% of all cancer patients, cancer cachexia is reported as the cause of death [15,69].

Frailty is another concept commonly associated with advanced disease and frequently conflated with cancer cachexia, but it is distinct in that frailty is more accurately described as a state of vulnerability to decline within the context of aging [30]. There are two approaches for clinical frailty assessment. One is the Fried frailty phenotype, which defines the condition as the presence of three or more of the following indicators: unintentional weight loss, hand grip strength weakness, diminished walking speed, exhaustion, and decreased physical activity level [30]. The second approach is termed the Rockwood frailty index and classifies frailty as the accumulation of age-related deficits [70]. The factors most associated with frailty are decreased gait speed and cognitive decline, as well as poor survival [71,72,73,74]. Once a patient has been clinically diagnosed with frailty, treatment goals center around improving the current deficits and preventing the onset of new ones [75,76,77]. If an adverse event linked to frailty, such as a mechanical fall, does occur, patients typically have reduced survival and do not regain full functional capacity [71,78]. Ultimately, frailty is closer to a mediating tool for connecting to outcomes rather than an etiologic concept that can be addressed with targeted interventions.

The most common complaint among advanced cancer patients is chronic fatigue. Commonly referred to as cancer-related fatigue (CRF), the diagnosis is defined by an excessive and persistent feeling of tiredness that impairs emotional, physical, or mental function [79,80,81]. CRF has many contributing causes, including systemic biology changes induced by cancer, cancer treatments such as chemotherapy or radiation therapy, or a combination of the two [56,80], and patient-reported fatigue can be directly linked to poor survival in non-small-cell lung cancer (NSCLC) [82]. Importantly, CRF can be mediated by the central nervous system (CNS), in which voluntary muscle activation is decreased, or by the peripheral nervous system (PNS), whereby the synchronicity of muscle contraction is reduced, leading to a decrease in strength [83,84]. Patients may experience isolated central or peripheral fatigue or both concurrently. Cognitive aspects of fatigue and cancer-related cognitive impairment are well documented in patients with CRF, exhibiting delayed recovery from cognitive interference and impairments in memory and attention [81,85,86]. Some studies suggest that CRF can be a prognostic indicator for future cognitive decline [86,87].

More broadly, cognitive decline is a complex condition in advanced cancer patients, influenced by natural aging, cancer treatments, metastasis, and systemic cancer biology [3,88,89,90]. Studies have found that objective assessments and subjective patient-reported measures of cognitive decline do not always align, but general symptoms include brain fog, memory failure, and a lack of awareness/attention [91]. Objective measures of cognitive impairment are more indicative of decline, and subjective measures tend to be influenced by psychological measures of depression and anxiety with increasing cognitive decline with increasing psychological symptoms [85]. The clinically relevant cognitive decline that impedes QoL can begin in ~30% of patients at 12 months prior to death and significantly increase to ~45% of patients in the last 0–3 months prior to death, suggesting that a sudden increase in cognitive impairments can be a prognostic indicator for advancing disease and survival [92]. Additionally, emotional distress, such as anxiety and depression, can be a strong co-factor for outcomes related to both cognitive decline and CRF [79,85].

As seen in Figure 1, each of the discussed facilitators of morbidity acts as a factor that contributes to functional decline. Additionally, each factor participates in a positive feedback loop that exacerbates other facilitators of morbidity and leads to exponential decline. Overall, functional capacity is a defining facet of health and is instrumental to both QoL and survival. It is possible that improving these functional deficits will not only improve quality of life but also improve treatment outcomes. Improving functional capacity requires knowledge of the biology underlying each of these facilitators of morbidity. The next section will focus on the contributing systemic biology of advanced cancer and will be followed by a comprehensive overview of current attempts to target this biology with pharmaceutical or exercise interventions for functional improvement.

## 3. Systemic Biology of Advanced Cancers: Extratumoral Mechanisms

Extratumoral mechanisms include any biology that is not directly tied to tumor mechanisms but could be influenced by tumors via paracrine or endocrine factors. Cancer cachexia has been studied extensively through the lens of the systemic biologic mechanisms, largely immune and metabolic, underlying advanced cancers. Functional assessments are not well defined in cancer cachexia biology studies and are still being developed. Currently, the most widely used tests include grip strength and locomotor activity, with a growing interest in muscle function in situ [93,94,95,96,97,98,99]. Unfortunately, most functional assessments are performed at the endpoint of a study, comparing morbid animals to sham; however, there has been a shift in the field to collect data longitudinally to better determine the early factors associated with cancer cachexia [94,95,97]. We will summarize the current known systemic mechanisms studied in cancer cachexia, which contribute to functional decline.

An increase in inflammatory cytokines in the blood is a well-reported event in cancer cachexia. The most frequently measured cytokines include IL-6, TNF-α, and TGF-β superfamily members, such as GDF-15 [100]. Cancer cachexia patients have been found to have significantly greater levels of circulating IL-6 and TNF-α, corresponding with decreased upper and lower body strength compared to healthy age-matched controls [101]. In mouse models, mice injected with adeno-associated viral vectors (AAV) expressing IL-6 and activin A, a member of the TGF-β superfamily, led to significant reductions in body weight, fat, and muscle mass, with the greatest effect when both cytokines were elevated [102].

Multiple mechanisms are suggested by which these inflammatory cytokines regulate muscle loss. One implicated pathway is the ubiquitin–proteosome pathway (UPP). This pathway is suggested to play a direct role in the muscle degenerative facet of cachexia in both animals and humans and is initiated when an increase in cytokines such as IL-6, IL-1β, and activin A triggers the downstream activation of UPP regulatory genes such as the muscle-specific E3 ligase MURF1 and atrogin-1. The increased activation of these genes then leads to protein degradation and muscle wasting [103,104,105,106]. In fact, studies have found that MURF1 activation alone is sufficient for muscle wasting, and a mouse MURF1 knockout model protects against skeletal muscle and fat wasting [107]. Mouse studies have also shown a significant decrease in muscle mass with elevated IL-6 [104,105]. In patients, high levels of IL-6, IL-1β, and IL-8 are significantly associated with PROMs of appetite loss and weakness [108]. However, it remains to be studied how UPP regulatory genes directly correlate with function in cancer cachexia in humans, and more work needs to be done to link function to cytokines at the human level.

Muscle biopsies from cancer patients with and without cachexia indicate that miRNAs that relate to myogenesis and inflammation may also have prognostic and predictive value [109]. Plasma levels from head and neck cancer patients with and without cachexia showed a decrease in miR-130a levels with an increase in TNF-α levels that was highly specific to individuals with cachexia, demonstrating potential as a cancer cachexia biomarker [110]. Although studies have not been carried out to determine the role these miRNAs may play in patient function or muscle loss, the high specificity suggests they may play a role in the pathophysiology of cancer cachexia.

In addition to muscle and adipose wasting, many other organs are negatively affected by increased cytokine production and contribute to systemic disease physiology. Liver dysfunction is one of the most studied outcomes of increased inflammatory cytokine circulation, and disrupted function leads to imbalances in glucose and insulin homeostasis, as well as steatosis and cholestasis [16,20,111]. Cytokine influx into the liver also triggers the hepatic inflammasome pathway in which IL-1β and the acute phase response proteins serum amyloid A and fibrinogen are released, causing increased blood lactate and subsequent increased resting energy expenditure [20,112,113,114]. Alterations in intestinal microbiota and gut permeability are also implicated with increased circulating IL-6, negatively impacting nutrient absorption and downstream metabolism [16,20]. Studies suggest the cytokines IL-1β and TNF-α play a role in the neuroendocrine regulation of appetite as increased levels of these cytokines in the brain are associated with a greater incidence of anorexia in mouse models, and inhibition of TNF-α in rats was shown to forestall anorexia [115,116,117].

The link between inflammation and the neural network that establishes appetite suppression in anorexia–cachexia syndrome has been an emerging area of focus in cachexia. Upregulated CNS-activating factors with growing interest in this area include GDF-15 and lipocalin 2 (Lcn2). GDF-15 is thought to be secreted by the liver and binds to the receptor GFRAL in the brainstem where upregulation modifies hunger by inducing nausea and emesis [118,119]. These symptoms result in a decrease in food intake and body weight. Therefore, it is not surprising that cancer cachexia patients experience an upregulation of GDF-15, and exciting advances have been made to inhibit the GDF15-GFRAL axis. Likewise, patients with pancreatic cancer have shown increased mortality correlated with an increase in Lcn2 expression. Lcn2 is derived in bone marrow and binds to melanocorticotropin 4 receptor (MC4R) in the hypothalamus, inducing anorexia. A mouse model that investigated the effects of both deletion and rescue of Lcn2 found protection and sufficient induction of anorexia–cachexia syndrome, respectively [120]. Other studies have found neural infiltrating neutrophils that express CCR2 in the velum interpositum of mice with pancreatic ductal adenocarcinoma (PDAC). When CCR2 expression was blocked or genetically deleted, there was a significant reduction in infiltrating immune cells, the expression of atrogenes, and loss of body weight [121]. Recent work in cachectic mice indicates a direct negative effect of these neuroendocrine mechanisms on cognitive function [115,121,122].

## 4. Pharmaceutical Interventions

Despite the wealth of pre-clinical work, only a handful of FDA-approved drugs are available for advanced stages of cancer, primarily via immune-based therapies. Treatments targeted toward metastasis have largely failed in clinical trials, with no change, worsened survival, or serious side effects, such as in metalloproteinase inhibitor (MPI) trials [123,124]. Conversely, one promising treatment for advanced cancer is tumor-infiltrating lymphocyte (TIL) therapy, specifically in metastatic melanoma, citing a 22% complete tumor regression in patients previously considered incurable [125]. However, TIL cannot be used universally, and non-responders have been found to have lower levels of CD27+ CD8 T-cells and shorter telomeres [125]. The low success with current targeted advanced cancer treatment options suggests a different approach to treatment, such as therapies that target the downstream systemic biology of cancer, are worth investigating.

While there are no FDA-approved treatments for cachexia, the current approach commonly used in “cachexia clinics” is multi-modal, including dietary optimization, pharmacologic interventions targeted toward nutrition impact symptoms, and physical activity/exercise [63]. In select studies, these clinics have been shown to improve patient weight loss and quality of life [126,127,128]. The use of anti-inflammatories within a multi-disciplinary intervention has been proposed, although their overall efficacy remains unclear [129]. In one study of indomethacin treatment, cancer patients increased median survival from 250 days to 510 days [130], but more recent studies have demonstrated that the role of NSAIDs in treating cachexia is unclear and further study is still needed [131]. Corticosteroids have demonstrated improved body weight and appetite in cancer cachexia patients; however, treatment is recommended to last no more than a few weeks due to accumulated toxicity and side effects with long-term use. For this reason, corticosteroid treatment is often limited to end-of-life care [132]. More broadly, the concept of a “cachexia clinic” remains rare worldwide, with the vast majority of cancer centers lacking any dedicated clinical personnel for muscle wasting-related care. This is in part due to the current lack of evidence for the specific elements of a multidisciplinary approach to cachexia [133].

Efforts to produce drugs specific for cancer cachexia mechanisms have included blocking IL-6 signaling with monoclonal antibodies (mAb), including Clazakizumab and Tocilizumab (Toci). Initial patient responses to Toci treatment were favorable in two cancer cachexia patients with elevated IL-6 as well, demonstrating an increase in performance scores and enabling further chemotherapy treatment [134]; however, clinical trials have not been completed to fully examine the effectiveness of Toci in cancer cachexia patients [100]. Likewise, Clazakizumab showed increased grip strength and decreased fatigue-related symptoms in patients with NSCLC, but further clinical trials have not been carried out [135]. At the same time, Toci already has FDA approval for the treatment of cytokine release syndrome (CRS), which is an acute toxic response to CAR T-cell therapy, suggesting that Toci therapy is successful in reaching its biological target during cancer [136,137]. However, CRS only requires a short-term duration of therapy, and further study is needed Into long-term durations of therapy in order to successfully treat IL6-mediated cachexia.

Among emerging nutrition impact symptom-targeted therapies, anamorelin has shown much promise [138]. Anamorelin is a selective ghrelin receptor antagonist that was shown to increase lean body mass (LBM) and body weight in a phase 2 clinical trial with cancer cachexia patients [139]. Phase 3 studies of anamorelin demonstrated weight gain, an increase in LBM, and the amelioration of anorexia–cachexia symptoms but no change in function or grip strength [140,141,142], which then led to a lack of approval. However, given the paucity of mechanistic evidence linking leptin–ghrelin signaling to physical function, the absence of any changes in physical outcomes is not unexpected and perhaps should have been excluded as a requirement for approval in a unimodal study design.

Pharmaceutical intervention data with a GDF-15 inhibitor mAb show promising results in mouse models. Along with increases in muscle mass, appetite, and food intake, mice also demonstrated improved functional abilities with wheel and treadmill running assessments compared with non-treated animals [143]. Moreover, control IgG-treated animal pairs fed with tumor-bearing GDF-15-GFRAL axis-impaired mice showed greater body mass improvement in the tumor group, suggesting a feeding-independent mechanism for GDF-15 in cancer cachexia [144]. Clinical trials are now in phase 2, following the phase 1 study of the GDF-15 inhibitor ponsegromab, which demonstrated safety and secondarily resulted in increased body weight, appetite, and functional assessment and QoL scores [145,146].

The MC4R antagonist TCMCB07 trialed in rats with cachexia demonstrated a significant reduction in the loss of each body weight, cardiac and skeletal muscle, and fat [147,148]. Further studies in cachectic canines showed increased body weight, body condition score, and QoL as described by owners [149]. Preliminary data in the phase 1 study of TCMCB07 revealed no adverse effects or contraindications in healthy individuals [150].

Altogether, the search for cancer cachexia pharmaceutical interventions has not yet been fruitful. The failure of individual agents to treat cancer cachexia may be tied to the heterogeneity of biological cachexia presentation between patients and over time [142]. Since patients do not have equally elevated levels of cytokines or serum inflammatory markers, a more precise approach should be taken on the level of individualized or precision medicine in the oncologic community to determine the most appropriate pharmaceutical intervention, as has been conducted on a limited basis in some reported cases. Within such an approach, clusters of inflammatory markers should be considered regarding prognostic outcomes and treatment success. Furthermore, a disease-modifying marker that acts as a proxy to test therapy mechanistic action would improve treatment accuracy and efficacy.

## 5. Nutritional Guidelines and Interventions

Within cancer cachexia clinics and individual oncologic care, nutrition is routinely addressed by physicians and/or registered dieticians to provide safe recommendations for dietary intake, as well as education about the macronutrient content of specific food groups [132]. The current European Society of Clinical Nutrition and Metabolism (ESPEN) guidelines established in 2021 for cancer patient nutritional intake vary depending on the patient’s presentation. Cancer patients with weight loss are recommended protein intake of 1–1.5 g/kg/day, supplementation with vitamins and minerals to recommended daily allowances, and, in the presence of insulin resistance, an increased ratio of energy from fat than carbohydrates. In cases with severe or worsening malnutrition due to inadequate oral intake, enteral or parenteral nutrition may be recommended [151].

Emerging opportunities for nutritional intervention include the ketogenic diet. A recent study with two cancer cachexia mouse models demonstrated that although a ketogenic diet alone decreases survival and tumor burden, a ketogenic diet plus dexamethasone decreases tumor burden and mRNA levels of E3 ligases, as well as significantly extends OS and PFS compared to normal-fed plus dexamethasone-treated mice [152]. These findings suggest that certain diets may help improve the effects of cancer cachexia; however, they may need to be augmented with medications or more advanced nutritional strategies. Current data for the use of special diets are limited, and more work is required to understand the clinical context in which diet and augmenting medications can be used.

## 6. Rehabilitation and Exercise Interventions

In 2018, The American College of Sports Medicine (ACSM) Roundtable convened to determine exercise frequency, intensity, time, and type (FITT) prescriptions based on available evidence for cancer patients. FITT prescriptions were developed with strong evidence for directly addressing the aforementioned facilitators of morbidity, including fatigue [153,154,155,156,157,158,159,160,161,162,163], QoL [164,165,166], and physical function [165,166,167]. Each of these health outcome prescriptions recommends a regimen of aerobic, resistance, or a combination of both 3x times per week for 6 to 12 weeks with specifications for the duration, intensity as a measure of HRmax or VO(2max), number of repetitions, and whether supervised or unsupervised shows the most benefit [168]. While these prescriptions are a large step toward incorporating physical activity as a treatment, limitations to implementation exist. For instance, this recommendation was based on studies that largely focused on one type of cancer: breast or prostate. Additionally, studies were performed in individuals undergoing or completing cancer treatment, meaning enrollment for advanced cancer patients was low and therefore may not translate well to this population.

Cochrane reviews for exercise treatment, specifically in advanced cancer and cachexia patients, state that although exercise is reliably safe and there is some evidence it can lead to decreased symptom burden, the heterogeneity of studies precludes determination as a recommended treatment choice [169,170,171,172]. Largely, these studies are small and implemented within individual institutions, resulting in a lack of consensus between clinical approaches, types of exercises, and exercise duration/frequency, all of which are needed to administer an exercise prescription. Within these differences lie important distinctions that are often lost: rehabilitation and exercise are separate concepts, and aerobic and resistance exercise have different values physiologically. Here, we will review these distinctions, as well as published data on their impact on facilitators of morbidity and function in advanced cancer patients. Table 1 reviews the specific literature discussed in this section, defining each study as rehabilitation, exercise, or a combination.

The goal of rehabilitation interventions is patient- and task-specific (such as ADLs, mobility, or higher-level activities/sports), while exercise is focused on general increases in either strength (resistance) or endurance (aerobic) [35] and is therefore tailored to specific physiological goals rather than functional independence goals. Many studies have been carried out with each of these approaches in cancer patients, and common outcomes among them are increased QoL and a decrease in fatigue symptoms [40,41,42,160,173,178,179,180,184,185]. Physical function improvements are less common and more dependent on details of the rehabilitation or exercise intervention, as well as its level of implementation and appropriately matched measured outcome. Most rehabilitation-based programs will administer tests that focus on functional independence to assess improvement, whereas exercise interventions will focus on symptom or performance/activity metrics [35].

There are two main exercise modalities. Aerobic exercise is focused on cardiorespiratory fitness, and the intensity level can be determined by the total percentage of maximal HR or maximal oxygen consumption (VO2max) maintained or met during the time of exercise [186]. Activities involve repetitive movement of large muscle groups, such as those used in swimming, running, and biking. In contrast, resistance training consists of exercises focused on one muscle or muscle group at a time. Intensity is determined by the weight used during exercise as a percentage of the maximum weight that can be lifted during one repetition (1 RM). The main goal of resistance exercise is to induce muscle hypertrophy and increase overall muscle strength and endurance [187]. As seen in Table 1, clinically relevant improvements in cancer patients can be obtained with regimens for either aerobic or resistance exercise or in a mixed regimen including both exercise types.

The best evidence for physical activity-related interventions in advanced cancer includes a combination of resistance training specific to individuals and muscle groups. A large study of both aerobic and resistance training found an increase in physical performance defined by an increase in the shuttle walk test (SWT) and HGS [174]. Another study of inoperable lung cancer patients with 90 min of resistance exercise twice a week and 30 min of increasing walking intensity 3 times a week found an increase in 6MWT, Vo(2peak), muscle strength, and emotional QoL [178]. When compared against each other, not surprisingly, resistance training and not aerobic training uniformly improved muscular strength and/or endurance as defined by 6MWT, staircase walking, sit-to-stand, and TUG [160,176,177,180,181,183,188], while targeted muscle group exercises have also shown positive results for improved mobility and ADL measured via PROMs [42,173]. Studies examining the benefits of higher-intensity rehabilitation interventions for advanced cancer or cachexia are limited, although one recent study showed that patients with greater cachexia severity can still improve their functional independence in the inpatient rehabilitation setting [29].

The underlying cause for improved fatigue and QoL with exercise, in both individuals with and without illness, has perplexed researchers for decades. There is a large interest in the anti-inflammatory effects of exercise, but a clear mechanism is not yet understood. Interestingly, although IL-6 and NFκB are elevated in cancer cachexia [104,189,190,191], rodent models and human studies have shown improved cancer cachexia and survival with exercise, but with unexpected key mechanistic results, such as increased IL-6 and NFκB signaling [19,192,193,194]. However, investigation of cytokine cascades in both sepsis and acute exercise (not specified as resistance or aerobic) revealed a key difference in that pro-inflammatory cytokines TNF-α and IL-1β are present first in sepsis before the release of anti-inflammatory cytokines IL-6, TNF-R, IL-10, and IL-1ra [194]. In contrast, no TNF-α or IL-1β are released before the presence of anti-inflammatory cytokines in exercise [194]. Considering elevated levels of TNF-α and IL-1β are found in advanced cancer and cachexia patients, these results suggest that these cytokines may be more interesting therapeutic targets [195,196]. Furthermore, this inflammatory insight underscores the potential for more investigation into the variability of cytokine expression in cancer patients in the context of physical activity, exercise, or rehabilitation.

Overall, the current use of rehabilitation and exercise for advanced cancer and cachexia is limited due to outcome variabilities and discordant implementations between study groups; however, reasonable inferences can be made toward a potentially effective clinical training regimen. As the functional capacity of cancer patients varies widely between individuals, the first step is to implement exercises that are specific to individual functional levels. Patients with a disability should receive rehabilitation regimens aimed at increasing functional independence rather than focusing on physiologic goals. Meanwhile, for those without substantial physical impairments or disabilities, exercise should be geared toward specific physiologic goals, such as improving objective muscle strength and coordination or cardiopulmonary fitness (e.g., VO_2_max). A mixed approach would likely be ideal given the distinct health benefits of both aerobic and resistance modalities, but it has not been specifically studied with esither cancer or cachexia-specific outcomes, unlike sarcopenia, where exercise recommendations have been more firmly established [197]. Further studies are needed to test this tiered, tailored approach in specific cancer populations and to understand the details of the rehabilitation and exercise prescriptions that will be needed for both effective improvement in functional outcomes but also easy implementation.

Finally, the overlap in cytokines released from exercise and overexpressed in cancer and cachexia muddles the deterministic properties of these markers for successful exercise intervention. A clear biological marker for the success of rehabilitation and exercise intervention is greatly needed to help identify the most effective treatments and establish a consensus with which a specific prescription can be given. Studies should also carefully match outcomes to the specific type of physical intervention being offered for patients, as these decisions likely have an understated role in the likelihood of ultimate success in future clinical trials.

## 7. Conclusions and Future Directions

There is currently a lack of FDA-approved pharmacological agents and therapies for advanced cancer patients, and morbidity and mortality rates remain high. While many tumor biology-centered approaches have been tried and failed, some mechanisms for treating within the realm of advanced cancer downstream systemic biology have proven successful in reducing the morbidity burden, and new trials are revealing potential pharmacological agents that may increase survival, such as those targeting the cytokine GDF-15. In the meantime, resistance exercise may be a promising treatment to increase function and QoL in advanced cancer individuals. Current research suggests more work needs to be carried out to determine the most effective treatment for this disease, including dosage and intensity. It is evident more collaboration is needed to ensure enough large trials are conducted to reasonably infer the success of treatment intervention.

Future studies should focus on determining the most effective doses and intensities of rehabilitation and exercise interventions and, importantly, define the most important outcomes which determine intervention success. More work needs to be carried out on the interplay between cytokines and inflammatory agents in advanced cancer to determine how pharmaceutical interventions may impact treated individuals. Furthermore, once a better platform for these markers is established, measurable changes in these markers over the treatment timeline should be adopted as an indication of treatment response and overall success. Lastly, mechanisms for integrating precision-based treatment of cachexia or advanced cancer, which may also include multi-modal approaches, should be tested at the pre-clinical and early-phase clinical levels.

## Figures and Tables

**Figure 1 cancers-16-00360-f001:**
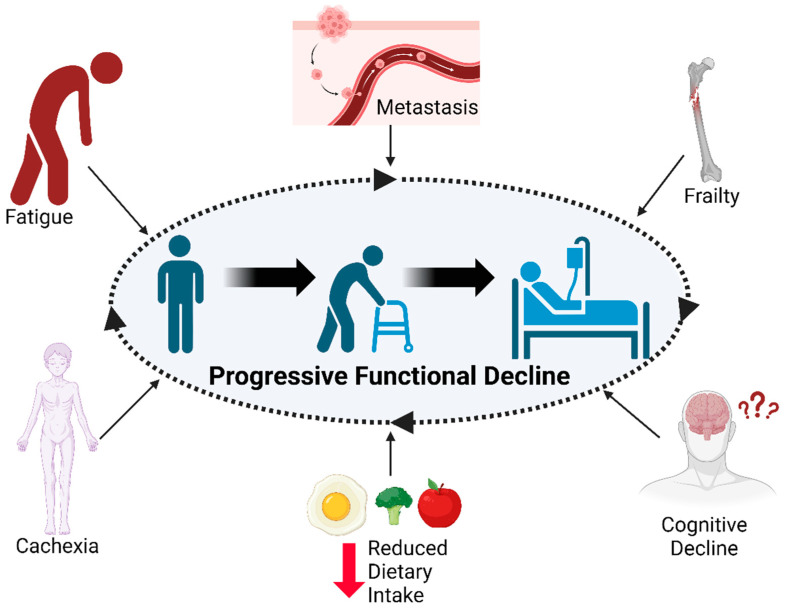
Factors which contribute to progressive functional decline. Advanced cancer facilitators of morbidity act as driving factors for progressive functional decline. The presence of functional decline exacerbates these contributing factors, and a decline in each factor may lead to a consequent decline in others.

**Table 1 cancers-16-00360-t001:** Published rehabilitation or exercise interventions in advanced cancer patients.

Cancer Type	Length of Time	Intervention Type (Exercise or Rehabilitation?)	Routine	Improved Outcome Measures	References
Stage IV lung (*n* = 34) and colorectal (*n* = 32) cancer	8 weeks	Exercise	4×/week incremental 20 min brisk walks REX: 5 exercises, 10–15 reps	Mobility, fatigue, and sleep	Cheville et al., 2013 [173]
Mixed incurable metastatic cancers with life expectancy < 2 years (*n* = 121)	8 weeks	Exercise	2×/week 50–60 min AEX: 10–15 min REX circuit: 6 stations, 2 min on, 1 min off	Shuttle walk test (SWT) and HGS	Oldervoll et al., 2011 [174]
Mixed recurrent advanced or metastatic cancer (*n* = 46)	12 weeks	Exercise	30 min walking on alternate days	None	Tsianakas et al., 2017 [175]
Stage IIIb or IV NSCLC or extensive small cell lung cancer (*n* = 114)	6 weeks	Exercise	2×/week supervised REX and AEX 3×/week home walking	Peak VO2, 6MWT, and 1 RM	Quist et al., 2015 [176]
Advanced NSCLC (*n* = 40)	8 weeks	Exercise	3 days/week supervised, 2 day/week independent REX and AEX	6MWT and dynamometer muscle strength	Kuehr et al., 2014 [177]
Stage IIIb or IV NSCLC or extensive small cell lung cancer (*n* = 23)	6 weeks	Exercise	2×/week supervised AEX and REX 3×/week home walking	Peak VO2, 1 RM, and emotional QoL	Quist et al., 2012 [178]
Stage IV breast cancer (*n* = 38, 16 intervention)	12 weeks	Exercise	3×/week video-guided seated REX and stretching	Slower decline in total and physical well-being Less increase in fatigue	Headley et al., 2004 [179]
Mixed advanced cancers (*n* = 34)	6 weeks	Exercise	2×/week supervised REX circuit training	6MWT, timed sit-to-stand, physical fatigue, and emotional QoL	Oldervoll et al., 2006 [180]
Prostate cancer (*n* = 16)	12 weeks	Exercise	3×/week 12–15 min eccentric resistance cycling	6MWT and isometric knee extension strength	Hansen et al., 2009 [181]
Mixed advanced cancers (*n* = 115)	8 sessions	Rehabilitation	3×/week 90 min PT for trunk and lower extremity	Physical well-being	Cheville et al., 2010 [182]
Stages III and IV lung cancer (*n* = 46)	Duration of 3 chemotherapy cycles	Rehabilitation	3–4×/week REX: 50% capacity 10 reps, 3 sets with resistance bands 5×/week AEX: 6 min walk moderate intensity 2 min staircase walking	Cognitive function, physical function, staircase walking, and 6MWT	Henke et al., 2014 [183]
Inpatient advanced cancer patients (*n* = 250)	Length of hospital stay	Rehabilitation	Minimum of 900 min of therapy/week, including PT, OT, and SLT	Motor and cognitive FIM scores	Roy et al., 2023 [29]

Individual studies are classified by cancer type and stage (when available), duration, intervention type, routine, and improved outcomes after intervention. REX: resistance exercise; AEX: aerobic exercise; 1 RM: one repetition maximum; PT: physical therapy; OT: occupational therapy; SLT: speech and language therapy; FIM: functional independence measure.

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
