# Peer review of "Harnessing the Systemic Biology of Functional Decline and Cachexia to Inform more Holistic Therapies for Incurable Cancers"

_cancers, 2024, doi:10.3390/cancers16020360_

Round 1
Reviewer 1 Report
Comments and Suggestions for Authors
This narrative review written by Willbanks focuses on functional decline in patients with cancer cachexia and gives future perspectives to understanding, diagnosis, and therapeutics for incurable cancer. It is well organized and readable for those who are interested in cancer patients. There are several comments.
1. Subheading: There are two “chapter 2”. The chapter titled “Systemic Biology of Advanced Cancers: Extratumoral mechanisms” may be chapter 3, and the following chapters will be changed, ending in chaper “6” Conclusions and Future Directions.
2. Line 93 page 2: “Among the first signs of advanced cancer is decreased dietary intake.” Is it grammatically correct?
3. Line 120 page 3: Frailty is a concept of a state of vulnerability to decline, as shown by the authors. It is mainly due to the age-associated mechanism rather than to the effect of diseases, as shown in Figure 1, in which the authors distinguish cachexia from frailty. It may be better to explain more clearly that frailty is a concept of age-associated state mainly in the field of geriatrics in this chapter.
4. Line 145 page 3: “Cognitive fatigue” Is it a well-known medical term? May it be better with quotation marks?
5. Line 178 page 5: “mechanisms” may be better the one capitalized.
6. Line 246 page 6: “Pharmaceutical interventions” Is there any need to show the use of corticosteroids, which are included in some cancer cachexia guidelines.
7. Is there any room for nutritional interventions, which are also included in some cancer cachexia guidelines.
Author Response
Please see attachment
This narrative review written by Willbanks focuses on functional decline in patients with cancer cachexia and gives future perspectives to understanding, diagnosis, and therapeutics for incurable cancer. It is well organized and readable for those who are interested in cancer patients. There are several comments.
We appreciate your positive comment and address concerns below.
Subheading: There are two “chapter 2”. The chapter titled “Systemic Biology of Advanced Cancers: Extratumoral mechanisms” may be chapter 3, and the following chapters will be changed, ending in chaper “6” Conclusions and Future Directions.
A: Thank you for catching this! It is now updated.
- Line 93 page 2: “Among the first signs of advanced cancer is decreased dietary intake.” Is it grammatically correct?
A: We believe this is grammatically correct and conveys our message.
- Line 120 page 3: Frailty is a concept of a state of vulnerability to decline, as shown by the authors. It is mainly due to the age-associated mechanism rather than to the effect of diseases, as shown in Figure 1, in which the authors distinguish cachexia from frailty. It may be better to explain more clearly that frailty is a concept of age-associated state mainly in the field of geriatrics in this chapter.
A: Thank you for this suggestion. We addressed this comment in line 122.
- Line 145 page 3: “Cognitive fatigue” Is it a well-known medical term? May it be better with quotation marks?
A: Cognitive fatigue was not directly stated in our reference articles, so we changed this term to “cognitive aspects of fatigue in line 145 to better convey the message from those articles.
- Line 178 page 5: “mechanisms” may be better the one capitalized.
A: We made this change in the manuscript-thank you!
- Line 246 page 6: “Pharmaceutical interventions” Is there any need to show the use of corticosteroids, which are included in some cancer cachexia guidelines.
A: Thank you for this suggestion. We included information about corticosteroids in lines 275-279 of the revised manuscript.
- Is there any room for nutritional interventions, which are also included in some cancer cachexia guidelines.
A: Thank you for this suggestion. We inserted a new section titled “Nutritional Guidelines and Interventions” at line 332 before the exercise section.

Reviewer 2 Report
Comments and Suggestions for Authors
The authors propose an interesting aspect of tumor management.
Physical activity should definitely be proposed, I have some points to evaluate better:
- Studies are reported, but it would be interesting to propose a possibly effective training scheme
- A distinction should be made between aerobic, strength, and mixed training, perhaps proposing hybrid schemes as could be the case for sarcopenia (10.3389/fspor.2022.950949)
- A hint on miRNAs would be interesting if there could be a possible regulation and possible benefits
- It would be interesting to evaluate the combination with dietary regimes such as ketogenic
Comments on the Quality of English LanguageJust few
Author Response
Please see attachment
The authors propose an interesting aspect of tumor management.
A: Thank you for your feedback. Please see our responses to each comment below.
Physical activity should definitely be proposed, I have some points to evaluate better:
- Studies are reported, but it would be interesting to propose a possibly effective training scheme
A: Thank you, this has been included at the end of the exercise section beginning at line 440. Our recommendation/proposal remains somewhat vague due to lack of evidence to support specificity.
- A distinction should be made between aerobic, strength, and mixed training, perhaps proposing hybrid schemes as could be the case for sarcopenia (10.3389/fspor.2022.950949)
A: Thank you, we included a new paragraph beginning line 396 to address this. Of note, in table 1 we already had identified which studies support aerobic, resistance, or mixed training in the “routine” column.
- A hint on miRNAs would be interesting if there could be a possible regulation and possible benefits
A: Thank you for the suggestion. A paragraph about miRNAs has been inserted at line 213.
- It would be interesting to evaluate the combination with dietary regimes such as ketogenic
A: Thank you for this suggestion. We inserted a new section titled “Nutritional Guidelines and Interventions” at line 332 before the exercise section.
